# Comparison of the Effects of Chemical Topping Agent Sprayed by a UAV and a Boom Sprayer on Cotton Growth

**Zechen Dou** [1], **Zhihao Fang** [1], **Xiaoqiang Han** [1,*], **Yapeng Liu** [1], **Li Duan** [1], **Muhammad Zeeshan** [1] **and Muhammad Arshad** [2]

1. Key Laboratory of Oasis Agricultural Pest Management and Plant Protection Resources Utilization, College of Agriculture, Shihezi University, Shihezi 832002, China; 20202112050@stu.shzu.edu.cn (Z.D.); fangzhihao@stu.shzu.edu.cn (Z.F.); 20212112042@stu.shzu.edu.cn (Y.L.); 20202012073@stu.shzu.edu.cn (L.D.); maharzeeshan218@gmail.com (M.Z.)
2. Department of Entomology, University of Sargodha, Sargodha 40100, Pakistan; makuaf@gmail.com
* Correspondence: hanshz@shzu.edu.cn; Tel.: +86-188-0993-1417

**Abstract:** In order to improve the spraying efficiency of cotton chemical topping agents, a field experiment was conducted to compare the effects of 250 g·L$^{-1}$ mepiquat chloride aqueous solutions sprayed by a T30 unmanned aerial vehicle (UAV) and a boom sprayer. The cotton agronomic parameters, pesticide utilization rate, yield, and fiber quality were determined. The results showed that the pesticide utilization rates of the T30 UAV and the boom sprayer were 48.71% and 30.37%, respectively. After 10, 20, 30, and 40 days of T30 UAV spraying, the plant height inhibition rates of the two varieties of cotton were 12.97%, 13.78%, 20.91%, and 26.14% for Xinluzao 52 and 8.64%, 13.37%, 14.72%, and 18.03% for Huiyuan 720. After spraying with a boom sprayer, the plant height inhibition rates were 7.94%, 11.13%, 18.23%, and 23.69% for Xinluzao 52 and 6.09%, 9.98%, 11.78%, and 15.14% for Huiyuan 720 after 10, 20, 30, and 40 days of spraying. The T30 UAV spraying chemical topping agent was shown to have a significant beneficial effect on the utilization rate of the pesticides, as well as the agronomic properties, yield, and the fiber quality of the cotton.

**Keywords:** cotton; chemical topping agent; DPC; unmanned aerial vehicle; pesticide utilization rate





## 1. Introduction

The full mechanization of cotton is the best way to simplify its cultivation. Xinjiang is a high-quality cotton production base in China. In 2021, the planting area and yield of cotton in Xinjiang accounted for 67.90% and 89.50% of the totals in China, respectively [1]. Topping cotton is a key step in the cultivation of the crop and its management. Timely topping can change cotton from the period of vegetative growth to that of reproductive growth and increase the number and weight of cotton bolls. However, the traditional method of cotton topping by hand has high labor requirements and a low efficiency, hindering the full mechanization of cotton [2,3].

Topping cotton with chemical pesticides is necessary in order to realize the whole-process mechanization of cotton planting. The use of a chemical topping agent can significantly regulate the vegetative growth of cotton and make the plant type more compact. It can also increase the number of bolls by increasing the number of fruit branches. However, the number of fruit branches is equivalent to that of artificial topping and has no significant effect on the yield [4,5]. After spraying plants with N, N-dimethylpiperidinium chloride (DPC, 750 mL·hm$^{-2}$), the application of an appropriate amount of topdressing nitrogen fertilizer (300 kg·hm$^{-2}$) can improve the photosynthetic performance of cotton and promote the distribution of photosynthetic products to reproductive organs on the basis of increasing the dry matter accumulation [6]. The application of multiple treatments of acetaminophen to a cotton crop at specific intervals can reduce the cost of production while having no negative impact on yield [7]. A single application of a chemical topping agent can also result

in good dry matter accumulation and a good bud boll distribution ratio [8], suggesting that chemical topping can be used instead of manual topping. Previously, researchers have studied the effects of the use of a chemical topping agent on the agronomic characteristics, yield, and quality characteristics of cotton with different spraying times, irrigation amounts, planting densities, and effective component contents [9–13]. However, chemical topping is mainly sprayed by a boom sprayer with a locomotive tractor/vehicle. In the process of pesticide spraying, the vehicle rolls the cotton plant, drags the fruit branches, rolls the field, etc., causing damage to the crop and ultimately affecting the yield.

The low-altitude operation of a plant protection unmanned aerial vehicle (UAV) has the advantages of causing no damage to cotton, providing a high pesticide utilization rate, and saving water. They have been widely used for pest control in cotton fields [14,15]. On small size mountain vineyards and hilly areas, spraying pesticides with UAVs showed higher efficiency. Sarri found that the working capacity of an UAV was 2-fold that of a sprayer gun, and 1.6-fold that of a knapsack sprayer [16,17]. These results showed that UAVs spraying pesticide have a more efficient spraying efficiency. At present, there are few reports on the spraying of chemical topping agents by UAV. Therefore, it is of value to explore the feasibility of the application of a chemical topping agent through spraying by a UAV. It was found that UAV spraying at 98% DPC soluble powder (SP) twice displayed similar effects on cotton plant height compared to spraying with a boom sprayer [18]. After spraying 98% DPC SP twice with a UAV (3WQF125-16), the plant height and fruit node growth of cotton were significantly lower than those of the control, showing a good effect on regulating cotton growth [19]. Cotton chemical topping agents with DPC, flumetralin, and chlormequat chloride as the main components have been widely used [20]. However, in the early years of the application of this technology, chemical topping agents needed to be sprayed twice with a DPC to effectively inhibit the growth of the cotton. With the optimization of formulations and the improvement of the spraying method used, nowadays it is only necessary to spray a chemical topping agent once to achieve the desired results. As different cotton cultivars differ in their sensitivity to DPC, two local cultivars were used for comparison. For this study, field experiments were conducted to analyze and compare the effects of chemical topping agents on the growth of cotton varieties, including Xinluzo 52 and Huiyuan 720, sprayed by a UAV and a boom sprayer. The pesticide utilization rate, agronomic characteristics, yield, and fiber quality are compared and analyzed in order to provide guidance for improving the effect of spraying a chemical topping agent on cotton.

## 2. Materials and Methods

### 2.1. Field Plots

This study was conducted in 2021 at two sites. The first site was Beiquan town, Xinjiang Production and Construction Crops, Shihezi, (44°18′44″ N, 86°3′25″ E), Xinjiang, China (Figure 1 and Table 1). The spraying was conducted on 19th July 2021, the average speed of the wind was 1.03 m·s$^{-1}$, the relative humidity was 42.90%, and the temperature was 28.93 °C (Kestrel 5500, Nielsen-Kellerman, Boothwyn, PA, USA). The second site was the Teaching Experimental Field of Shihezi University, Xinjiang Production and Construction Crops, Shihezi, (44°19′27″ N, 85°58′52″ E), Xinjiang, China. The spraying was conducted on 14 July 2021, the average speed of the wind was 1.06 m·s$^{-1}$, the relative humidity was 45.94%, and the temperature was 30.13 °C (Kestrel 5500, Nielsen-Kellerman, Boothwyn, PA, USA). Cotton had been planted continuously for many years in both experimental plots, and the soil fertility was at a medium level. The conventional wide and narrow row cotton sowing mode was adopted, with 6 rows of 1 film and a spacing of 66 + 10 cm. Drip irrigation with one film and two tubes was used throughout whole growth period. The planting mode, growth period, fertilization level, irrigation, conventional chemical topping agent spraying time, and dose of each treatment were kept the same.

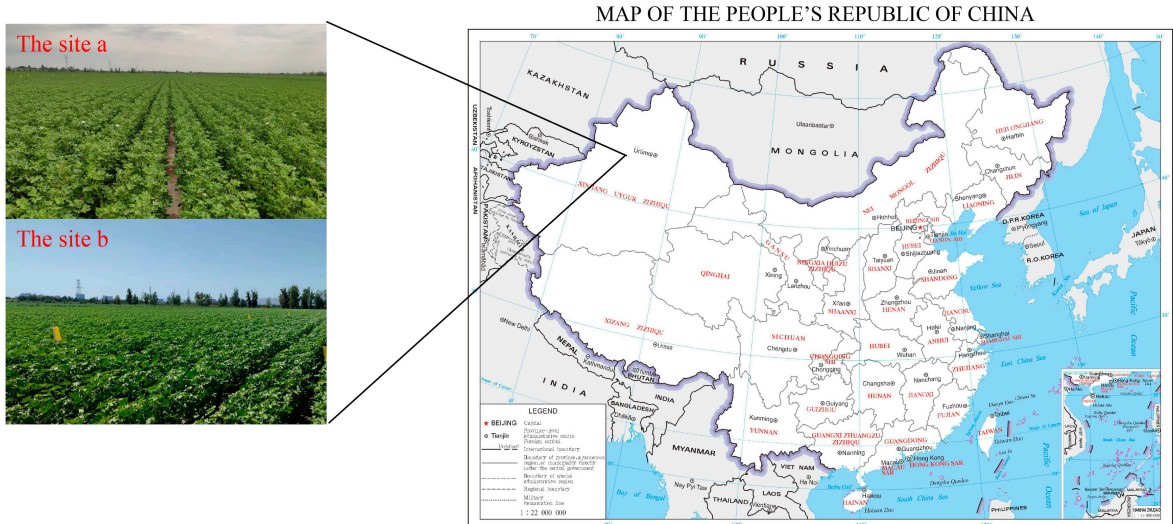

**Figure 1.** The graphical map of the study area.

**Table 1.** Planting overview of the test sites.

| Test Site | Cotton Varieties | Sowing Date | First Watering Time | Topping Time | Planting Density (Plants·ha$^{-1}$) |
|---|---|---|---|---|---|
| 1 | Xinluzao 52 | 19 April 2021 | 22 April 2021 | 14 July 2021 | 200,000 |
| 2 | Huiyuan 720 | 15 April 2021 | 24 April 2021 | 19 July 2021 | 195,000 |

### 2.2. Test Materials

Test reagent: 85% Allura Red was sourced from Zhejiang Jigaode Pigment Technology Co., Ltd., Longgang, China; 250 g·L$^{-1}$ DPC aqueous solution (AS) was sourced from Jiangsu Runze AgroChemical Co., Ltd., Changzhou, China; and a Dajiang T30 UAV was sourced from Shenzhen Dajiang Innovation Technology Co., Ltd., Shengzhen, China. The volume of the tank was 40 L, and it had dimensions of 2858 mm × 2685 mm × 790 mm (length × width × height) (arm deployment, paddle deployment), 6 rotors, and 16 SX11001VS nozzles. The operation parameters of the T30 UAV were input by the intelligent handheld terminal, and carrier phase difference technology was used for flight accurate positioning. During the test, the route spacing of the T30 UAV was 5 m (Figure 2), the flight speed was 5 m·s$^{-1}$, and the flight altitude was 2 m (from the ground). A boom sprayer was used with John Deere954 tractors (John Deere, Moline, IL, USA) as the traction main body and a farmer's self-assembling sprayer. The volume of the tank was 1000 L, with dimensions of 2400 mm × 1920 mm × 3150 mm (length × width × height), a working speed of 540 r·min$^{-1}$, a liquid pump flow of 100 L·min$^{-1}$, 32 fan-shaped nozzles, and a spray width of 12 m (length of boom). After measuring the nozzle flowrate at the left, middle and right positions of the boom three times, the flowrate of a single nozzle was 1.37 L·min$^{-1}$, and the travel speed was maintained between 4.2 and 4.5 km·h$^{-1}$. The boom height from the ground was 60 cm and the nozzle model was a fan-shaped nozzle (11003, Dongguan Huajue Spray Technology Co., Ltd., Dongguan, China).

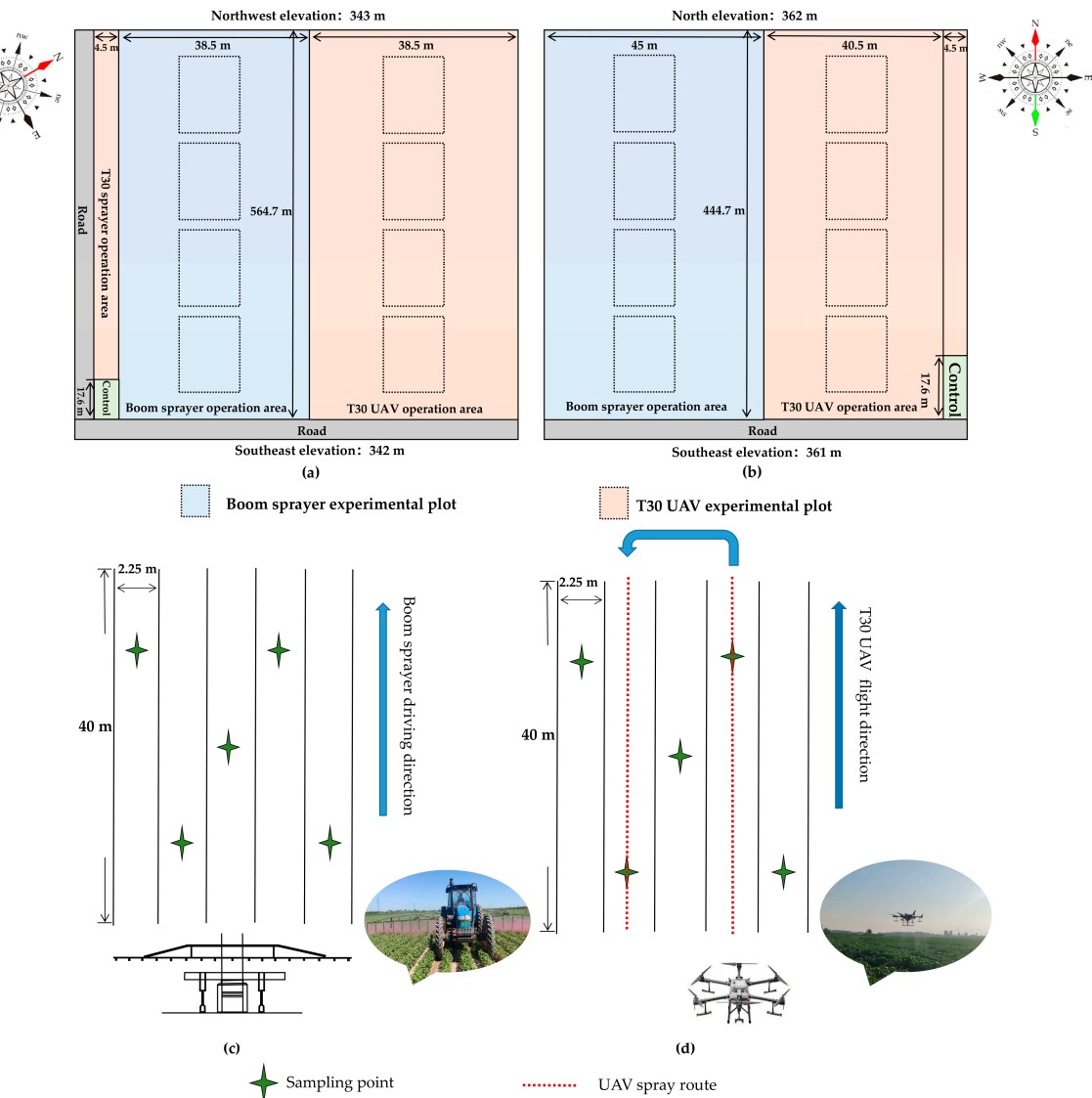

**Figure 2.** (**a**) Xinluzao 52 field plots, (**b**) Huiyuan 720 field plots, (**c**) boom sprayer experimental plot, and (**d**) T30 UAV experimental plot.

### 2.3. Test Method

#### 2.3.1. Test Treatment and Community Setting

The test site (a) extended from north to south, with an altitude of 362 m in the north and 361 m in the south. The test site (b) extended from northwest to southeast, with an altitude of 343 m in the northwest and 342 m in the southeast. The altitude difference between the two test sites did not exceed 1 m. In the two test sites, except for the control, the experimental plot was selected in the corresponding operation area for pesticide spraying with a T30 UAV and boom sprayer, with an experimental plot of 450 m$^2$, and each treatment was repeated 4 times (Figure 2a,b). Xinluzao 52 was planted in test site (a) with a planting area of 4.6 ha, of which 2.43 ha were sprayed by the T30 UAV and the rest by the boom sprayer. Huiyuan 720 was planted in test site (b) with a planting area of 4 ha, of which 2 ha were sprayed by the T30 UAV and the rest by the boom sprayer (Figure 3a,b). The two test sites were sprayed with 250 g·L$^{-1}$ DPC at once (the dosage was 750 g·ha$^{-1}$); the spray volume of the T30 UAV was 18 L·ha$^{-1}$ and the spray volume of the boom sprayer was 600 L·ha$^{-1}$ (Table 2). The average plant height of cotton in test site (a) was 52 cm, and the average plant height of cotton in test site (b) was 70 cm.

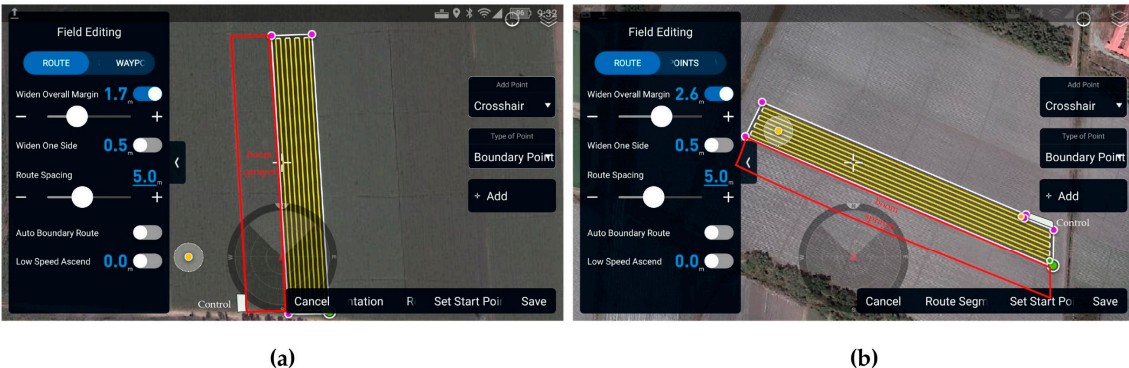

| | (a) | | | (b) | |

**Figure 3.** The spraying path for (**a**) Xinluzao 52 and (**b**) Huiyuan 720.

**Table 2.** Design of experiments.

| Treat | Cotton Varieties | Sprayed Equipment | Pesticide | Dosage (g·ha$^{-1}$) | Water Volume (L·ha$^{-1}$) |
|---|---|---|---|---|---|
| 1 | | T30 UAV | | 750 | 18 |
| 2 | Xinluzao 52 | Boom sprayer | 250 g·L$^{-1}$ DPC AS | 750 | 600 |
| 3 | | Control | - | - | - |
| 4 | | T30 UAV | | 750 | 18 |
| 5 | Huiyuan 720 | Boom sprayer | 250 g·L$^{-1}$ DPC AS | 750 | 600 |
| 6 | | Control | - | - | - |

### 2.3.2. Determination of Pesticide Utilization Rate
#### Sample Collection

Allure red was used as the indicator, with a fixed dosage of 450 g·ha$^{-1}$. It was added into the liquid of the chemical topping agent and was mixed thoroughly. Thirty minutes after spraying, the Z-shaped 5-point sampling method was adopted for sampling the test treatment (Figure 2c,d). Two cotton plants at each point were selected, cut with scissors along the base, and placed into a polyethylene bag (28 cm × 38 cm). Three experimental plots were used for each treatment and the sampling was repeated three times.

#### Allura Red Standard Curve

We accurately weighed 100 mg of allure red, dissolved it with 10 mL distilled water, and transferred it to a 100 mL volumetric flask. The volume was kept constant with distilled water to obtain 1000 mg·L$^{-1}$ of allure red mother liquor. Dilutions of 0.2, 0.5, 1.0, 2.0, 5.0, and 10.0 mg·L$^{-1}$ of allure red standard solution were prepared with distilled water. The absorbance value was measured at 514 nm by Tecan Infinite 200 PRO ELIASA (Tecan company, Männedorf, Switzerland), and the Allura Red concentration absorbance standard curve was obtained. The measured linear equation was Y = 0.0457 X + 0.0013 and the correlation coefficient was R$^2$ = 0.9999.

#### Pesticide Utilization Rate

The pesticide utilization rate was calculated according to the NY/T 3630.1-2020 standard using Formula (1):

$$D = \frac{\left(\rho_{smpl} - \rho_{blk}\right) \times F_{cal} \times V_{dil} \times \rho \times 10,000}{10^6 \times M \times N} \times 100 \tag{1}$$

where $\rho_{smpl}$ is the absorbance value of the sample; $\rho_{blk}$ is the absorbance value of the blank control; $F_{cal}$ is the slope value of the standard curve; $V_{dil}$ is the volume of eluent, in milliliters (mL); $D$ is the pesticide deposition and utilization rate of cotton (%); $\rho$ is the planting density (plant·m$^{-2}$); $M$ is the total amount of indicator per unit area applied (g·ha$^{-1}$); and $N$ is the number of sampled plants.

Agronomic Characters

Five points were selected on the diagonal line in each experimental plot, and ten cotton plants were marked at each point for a total of fifty plants. Cotton plant height, number of fruit branches, petiole internode ratio, fruit branch length (top first and top second), internode length (the internode length of the top first to the top second fruit branches), and number of buds and bolls were investigated 1 day before and 10, 20, 30, and 40 days after pesticide spraying (Table 3).

**Table 3.** Method of surveying the cotton plant type index and yield index.

| Number | Physiological Index | Survey Method |
|:---:|:---:|:---:|
| 1 | Plant height | Determining the height from the cotyledon node to the top of the main stem. |
| 2 | Number of fruit branches | Measuring the number of fruit branches of the main stem of the cotton plant. |
| 3 | Petiole internode ratio | Determining the ratio of the length of the petiole of the main stem of the first fruit branch and the second fruit branch to the length of the first internode on the fruit branch in the same node. |
| 4 | Fruit branch length | Measuring the length of the top first and top second fruit branches. |
| 5 | Internode length | Measuring the internode length of the top first to the top second fruit branches. |
| 6 | Number of buds and bolls | Counting the total number of buds and bolls. |

Note: Cotton plant height, number of fruit branches, petiole internode ratio, fruit branch length (top first and top second), internode length (the internode length of the top first to the top second fruit branches) measured with pocket ruler (Deli Group Co., Ltd., Ningbo, China).

### 2.3.3. Yield Characteristics and Fiber Quality

The cotton yield was measured after all the cotton bolls were opened and 100 cotton bolls from the canopies (upper, middle and lower layer) were randomly tagged and collected in each experimental area to determine the cotton yield characteristics and fiber quality. The number of bolls, single boll weight, and yield index were recorded. All the sampling bolls for measuring the fiber quality components and lint cotton yields were tested using a high-volume instrument (HVI-900A, Uster, Knoxville, TN, USA) in the Inspection and Testing Center of the Ministry of Agriculture and Rural Areas (Urumqi).

### 2.4. Data Statistics and Processing

Data were compared across different application rates using analysis of variance (ANOVA). By comparing the mean one-way ANOVAs, the least significant difference (LSD) was selected. The confidence interval was set to 95% and $p < 0.05$ was chosen to indicate a significant difference between the two groups. SPSS 18.0 was used for data processing and analysis, and the figures were made using SigmaPlot 12.5 and Origin 2021.

### 3. Results

### 3.1. Effects on Pesticide Utilization Rate

The results showed that the pesticide utilization rate of the T30 UAV for Xinluzao 52 was 48.71% (Figure 4), and this was higher than that of the boom sprayer (30.37%). Huiyuan 720 showed the same trend, and the pesticide utilization rate of the T30 UAV was 54.52%, higher than that of the boom sprayer (41.66%). This indicates that more DPC was left on top of the cotton after spraying with the T30 UAV than by the boom sprayer, which may have affected the topping.

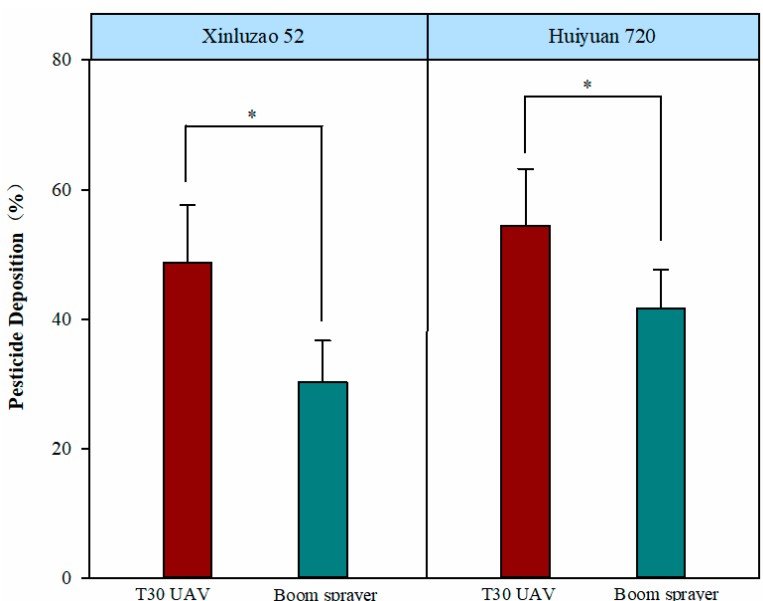

**Figure 4.** Effects on pesticide utilization rate. Note: * represents a significant difference compared with the other treatment ($p < 0.05$).

### 3.2. Effects of Chemical Topping Agent Sprayed by Two Different Sprayers on the Number of Cotton Bolls in the Fruit Branches and Plant Height

The same trend for the numbers of fruit branches was found in two cotton varieties (Figure 5). The number of fruit branches increased with the increase in treatment time and reached the maximum 20 days after pesticide spraying. In Xinluzao 52, the number of fruit branches was 10.44 after spraying pesticide with the T30 UAV and 10.49 when the boom sprayer was used. In Huiyuan 720, the number of fruit branches was 10.96 when the T30 UAV was used and 11.00 (numbers) when boom sprayer treatment was used, and there was no significant difference between two kinds of equipment (Xinluzao 52: $p = 0.886 > 0.05$, Huiyuan 720: $p = 0.133 > 0.05$).

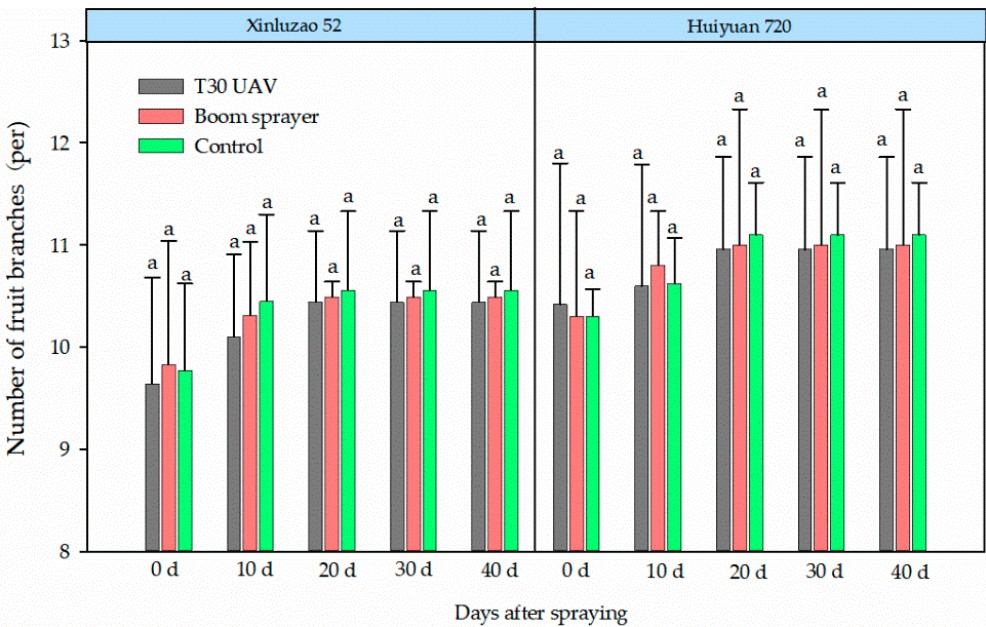

**Figure 5.** Effects of chemical topping agent sprayed by two sprayers on the number of cotton bolls in fruit branches. Different lowercase letters indicate significant differences between treatments at the 5% probability level ($p < 0.05$).

The T30 UAV and boom sprayer effectively inhibited the plant height of both cotton varieties (Figure 6). The plant height of Xinluzao 52 was increased by 6.28 cm after 10 days of pesticide spraying by the T30 UAV. Compared with the control, the inhibition rates 10, 20, 30, and 40 days after pesticide spraying by the T30 UAV were 12.97%, 13.78%, 20.91%, and 26.14%, respectively. After 10 days of spraying by a boom sprayer, the plant height was increased by 8.98 cm. Compared with the control, the inhibition rates 10, 20, 30, and 40 days after pesticide spraying by a boom sprayer were 7.94%, 11.13%, 18.23%, and 23.69%, respectively (Figure 6).

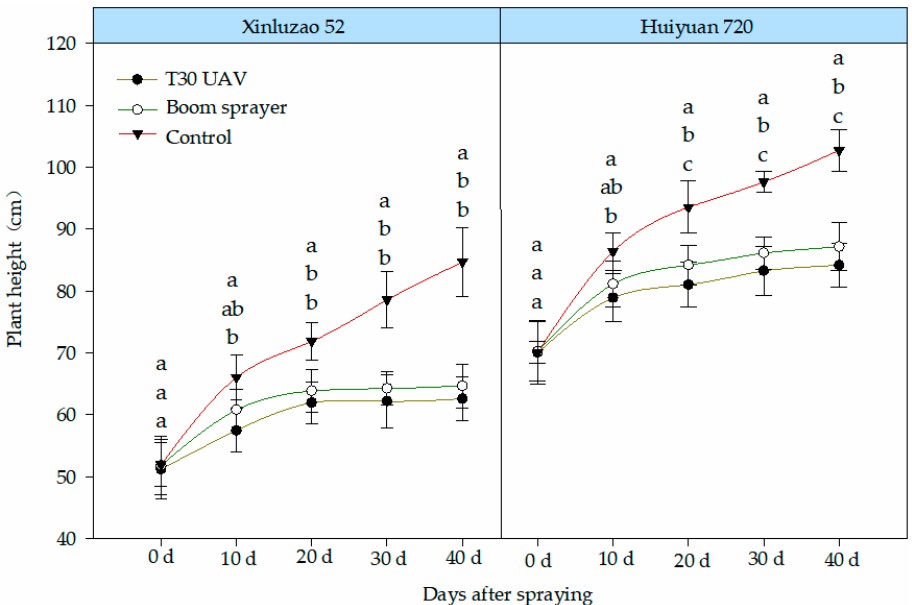

**Figure 6.** Effects of chemical topping agent sprayed by two sprayers on the cotton plant height. Different lowercase letters indicate significant differences between treatments at the 5% probability level ($p < 0.05$).

The plant height of Huiyuan 720 was increased significantly by the 10th day of pesticide spraying by the T30 UAV, and the rate of increase was 8.89 cm compared with the height before the pesticide was sprayed. Compared with the control, the inhibition rates 10, 20, 30, and 40 days after pesticide spraying by T30 UAV were 8.64%, 13.37%, 14.72%, and 18.03%, respectively. However, using a boom sprayer, the plant height was increased by 10.88 cm after 10 days of pesticide spraying. Compared with the control, the inhibition rates 10, 20, 30, and 40 days after pesticide spraying were 6.09%, 9.98%, 11.78%, and 15.14%, respectively, when a boom sprayer was used. Overall, with the passage of time, the daily increase in plant height after using a T30 UAV and a boom sprayer to spray chemical topping agents became smaller and smaller. The amount of chemical topping agent used in the T30 UAV was higher than that used by the boom sprayer (Figure 5). It is worth noting that the plant height sprayed by the T30 UAV was lower than that sprayed by the boom sprayer 10 days after spraying. After 10, 20, 30, and 40 days of spraying by the T30 UAV, the plant height inhibition rates of Xinluzao 52 and Huiyuan 720 were increased by 2.55~3.39% and 2.45~5.03%, respectively, compared with those of plants sprayed using the boom sprayer.

### 3.3. Effects of Chemical Topping Agent Sprayed by Two Sprayers on the Length of Fruit Branches

The results (Figures 7 and 8) showed that, among the two varieties used in this experiment, the growth levels of the top first fruit branch of Xinluzao 52 were 1.55, 2.60, 3.20, and 3.28 cm, respectively, after 10, 20, 30, and 40 days of pesticide spraying by the T30 UAV, while the growth levels of the top second fruit branch were 1.78, 2.26, 2.73, and 2.93 cm, respectively. Using a boom sprayer, the growth levels of the top fruit branch after 10, 20, 30, and 40 days were 2.15, 3.12, 3.22, and 3.28 cm, respectively, and the growth levels

of the top second fruit branch were 2.96, 3.39, 3.57, and 3.76 cm, respectively. After 10 days, the inhibition rates of the top first fruit branch and the top second fruit branch were 34.33% and 22.47%, respectively, when the T30 UAV was used. In the case of the boom sprayer, the inhibition rates of the top second fruit branch were 15.57% and 8.37% after 10 days (Figure 7).

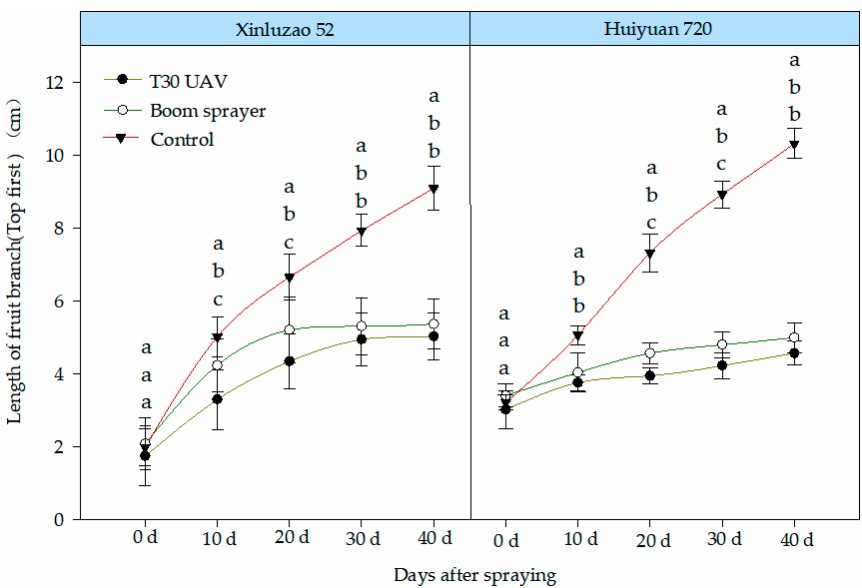

**Figure 7.** Effects of chemical topping agent sprayed by two sprayers on the length of fruit branches (top first). Different lowercase letters indicate significant differences between treatments at the 5% probability level ($p < 0.05$).

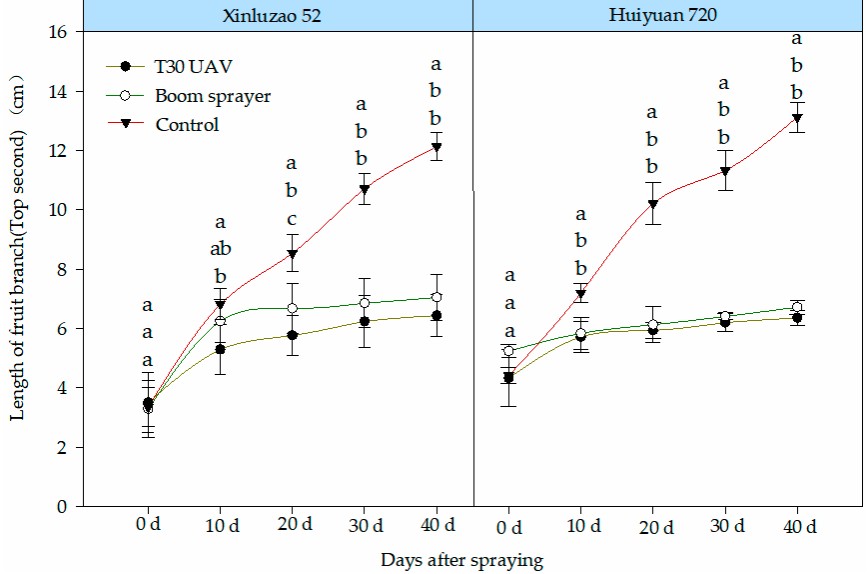

**Figure 8.** Effects of chemical topping agent sprayed by two sprayers on the length of fruit branches (top second). Different lowercase letters indicate significant differences between treatments at the 5% probability level ($p < 0.05$).

The growth levels of the top first fruit branch of Huiyuan 720 were 0.74, 0.93, 1.21, and 1.55 cm at 10, 20, 30, and 40 days after pesticide spraying by the T30 UAV, respectively, while the growth levels of the top second fruit branch were 1.39, 1.61, 1.87, and 2.03 cm, respectively. In the case of the boom sprayer, the growth levels of the top first fruit branch after 10, 20, 30 and 40 days were 0.63, 1.16, 1.39, and 1.59 cm, respectively, and the growth levels of the top second fruit branches were 0.60, 0.89, 1.17, and 1.48 cm, respectively. After

10 days of using a T30 UAV sprayer, the inhibition rates of the top first and top second fruit branches were 25.89% and 20.58%, respectively, and these increased to 55.77% and 51.60%, respectively, after 40 days. In the case of the boom sprayer, the inhibition rates of the top first and top second fruit branches were 20.36% and 18.92%, respectively, after 10 days of spraying, and these increased to 51.60% and 48.86%, respectively, after 40 days of pesticide spraying. Overall, the rates of inhibition of fruit branches achieved by the chemical topping agents sprayed by the T30 UAV were higher than those of the boom sprayer, especially for the inhibition of the top first fruit branch (Figure 8). It was found that the top first fruit branch and top second fruit branch sprayed by the T30 UAV were lower than those sprayed by the boom sprayer 10 days after spraying. After 10 days of spraying by the T30 UAV, the inhibition rates of the top first fruit branch from Xinluzao 52 and Huiyuan 720 were 18.76% and 5.53%, which were higher than the values achieved by the boom sprayer. The inhibition rates of the top second fruit branch were 14.10% and 1.66%, which were higher than the values achieved by the boom sprayer too. In general, the inhibition rates on the fruit branch sprayed with chemical topping agent by the T30 UAV were higher than those of the branches sprayed by the boom sprayer, especially for top first fruit branch.

*3.4. Effects of Chemical Topping Agent Sprayed by Two Sprayers on the Internode Length of the Main Stem*

The results showed that (Figure 9) the spraying of chemical topping agents by a T30 UAV and boom sprayer can effectively inhibit the internode length of the main stem and prevent the excessive growth of cotton. Two cotton varieties showed no significant differences in terms of the cotton internode length of the main stem after 10, 20, 30, and 40 days of using the T30 UAV sprayer. After 40 days of using the T30 UAV sprayer on Xinluzao 52, the inhibition rate of the internode length of the main stem was 29.25%, and in the case of the boom sprayer the rate was 28.27%. However, for Huiyuan 720, the inhibition rate of the internode length of the main stem was 30.34% when using the T30 UAV and 28.28% when using the boom sprayer after 40 days. After 40 days of pesticide spraying by the T30 UAV, the inhibition rates of the internode lengths of the main stems of Xinluzao 52 and Huiyuan 720 were increased by 1.28% and 2.06%, respectively, compared with those of the plants sprayed using the boom sprayer. Overall, the internode length of the main stem sprayed with a chemical topping agent using the T30 UAV showed a higher inhibition rate (Figure 9).

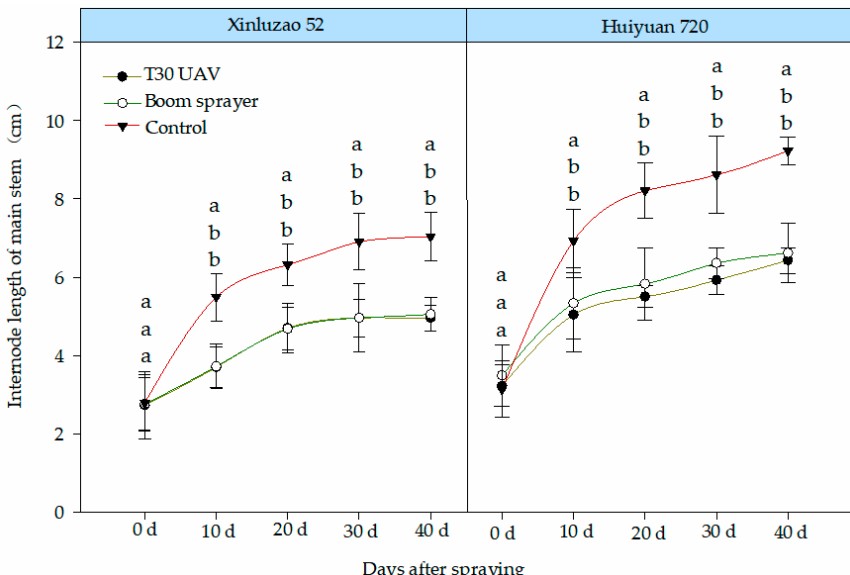

**Figure 9.** Effects of spraying a chemical topping agent with two sprayers on the internode length of the main stem. Different lowercase letters indicate significant differences between treatments at the 5% probability level (*p* < 0.05).

*3.5. Effects of Chemical Topping Agent Sprayed by Two Sprayers on the Cotton Petiole Internode Ratio*

The results show that (Table 4) the petiole internode ratio of the first fruit branch of Xinluzao 52 was 1.42, while that of the second fruit branch was 1.34 before the spraying. Before spraying Huiyuan 720, the petiole internode ratio of the first fruit branch was 1.60 and that of the second fruit branch was 1.75. At this time, the petiole internode ratio was about 1.60 and greater than 1.00, and the cotton showed a trend of rapid growth. For Xinluzao 52, on the 10th, 20th, 30th, and 40th days of pesticide spraying by the T30 UAV, the changes in the first fruit branch petiole internode ratio were −0.12, 0.00, 0.01, and −0.01, while the changes in the second fruit branch petiole internode ratio were −0.14, −0.01, 0.05, and 0. Using a boom sprayer, the changes in the first fruit branch petiole internode ratio were −0.19, −0.17, −0.11, and −0.09, while the changes in the second fruit branch petiole internode ratio were −0.12, −0.07, −0.05, and −0.06 (Table 4).

**Table 4.** Effects of chemical topping agent sprayed by two sprayers on the cotton petiole internode ratio.

| Cotton Varieties | Time after Treatment | Petiole Internode Ratio (First Fruit Branch) | | | Petiole Internode Ratio (Second Fruit Branch) | | |
|---|---|---|---|---|---|---|---|
| | | T30 UAV | Boom Sprayer | Control | T30 UAV | Boom Sprayer | Control |
| Xinluzao 52 | 0 d | 1.42 ab | 1.49 a | 1.34 b | 1.34 a | 1.35 a | 1.31 a |
| | 10 d | 1.30 a | 1.30 a | 1.18 b | 1.20 a | 1.23 a | 1.07 b |
| | 20 d | 1.42 a | 1.32 b | 1.15 c | 1.33 a | 1.28 a | 1.07 b |
| | 30 d | 1.43 a | 1.38 a | 1.14 b | 1.39 a | 1.30 a | 1.06 b |
| | 40 d | 1.41 a | 1.40 a | 1.18 b | 1.34 a | 1.29 a | 1.10 b |
| Huiyuan 720 | 0 d | 1.60 a | 1.69 a | 1.63 a | 1.75 a | 1.54 b | 1.64 a |
| | 10 d | 1.71 a | 1.34 b | 1.28 c | 1.57 a | 1.40 b | 1.27 c |
| | 20 d | 1.45 a | 1.32 b | 1.25 b | 1.50 a | 1.28 b | 1.17 c |
| | 30 d | 1.63 a | 1.53 b | 1.24 c | 1.52 a | 1.46 a | 1.29 b |
| | 40 d | 1.79 a | 1.60 b | 1.23 c | 1.75 a | 1.68 a | 1.25 b |

Note: Different lowercase letters indicate significant differences between treatments at the 5% probability level ($p < 0.05$).

The changes in the first fruit branch petiole internode ratio of Huiyuan 720 were 0.11, −0.15, 0.03, and 0.19, while the changes in the second fruit branch petiole internode ratio were −0.18, −0.25, −0.23, and 0.00 when a T30 UAV was used. In the case of the boom sprayer, the changes in the first fruit branch petiole internode ratio were −0.35, −0.37, −0.16, and −0.09, while the changes in the second fruit branch petiole internode ratio were −0.14, −0.26, −0.08, and 0.14. Overall, the petiole internode ratio of the first and second fruit branches of the two varieties decreased significantly after 10 days of pesticide spraying by a T30 UAV and a boom sprayer and rebounded after 20 days of spraying (Table 4).

*3.6. Effects of Chemical Topping Agent Sprayed by Two Kinds of Pesticide Equipment on Cotton Yield and Fiber Quality*

The results show that (Table 5) the ratio of harvested bolls could be effectively improved by spraying a chemical topping agent using the two sprayers. After 40 days of pesticide spraying by T30 UAV, the number of bolls in Xinluzao 52 increased by 1.83 compared with the control, while an increase of 1.28 bolls was achieved using the boom sprayer. After 40 days of pesticide spraying by the T30 UAV, the number of bolls from the Huiyuan 720 crop increased by 2.07 compared with the control, while an increase of 1.14 bolls was achieved when the boom sprayer was used (Table 5).

After spraying with a chemical topping agent, the mean length of the upper halves of the two varieties was significantly reduced, but no significant difference was found between the two sprayers. The breaking specific strength was significantly reduced, but no significant difference in Xinluzao 52 was found between the two sprayers. The application of pesticides by the two sprayers significantly increased the micronaire value. However, the micronaire value of Huiyuan 720 after spraying with the T30 UAV was 4.23, and this was not significantly different from that of the control. After spraying with a chemical

topping agent, the uniformity index and elongation at the break of the two varieties were significantly reduced. However, compared with the boom sprayer, the T30 UAV performed significantly better (Table 5).

**Table 5.** Effects of chemical topping agent sprayed by two sprayers on the cotton yield and fiber quality.

| Treatment | Cotton Varieties | Number of Bolls per Plant | Single Boll Weight(g) | Ratio of Harvested Bolls (%) | UHML (mm) | UNI (%) | MIC | STR (cN·tex$^{-1}$) | Elongation at Break (%) |
|---|---|---|---|---|---|---|---|---|---|
| 1 | | 9.13 a | 5.82 a | 53.54 a | 28.78 b | 83.80 b | 4.78 a | 29.73 b | 10.50 a |
| 2 | Xinluzao 52 | 8.58 b | 5.77 a | 45.77 b | 28.43 b | 83.07 c | 4.75 a | 29.08 c | 10.15 b |
| 3 | | 7.30 b | 5.10 b | 40.41 c | 30.15 a | 84.88 a | 4.35 b | 31.35 a | 10.70 a |
| 4 | | 10.10 a | 5.74 a | 71.09 a | 29.85 b | 84.88 b | 4.23 b | 31.20 b | 10.58 b |
| 5 | Huiyuan720 | 9.17 b | 5.70 a | 52.50 b | 29.77 b | 83.98 c | 4.45 a | 30.80 b | 10.08 c |
| 6 | | 8.03 c | 4.99 b | 45.87 c | 30.80 a | 85.55 a | 4.10 b | 32.60 a | 11.00 a |

Note: UHML—upper half mean length; UNI—uniformity index; MIC—micronaire; STR—strength. Self-comparison within the same cotton variety. Different lowercase letters indicate significant differences between treatments at the 5% probability level ($p < 0.05$).

## 4. Discussion

In this study, the pesticide utilization rate, agronomic characteristics, yield, and quality indexes of cotton were studied after the application of a chemical topping agent by two types of sprayers. The research showed that the T30 UAV had a higher pesticide utilization rate than a traditional boom sprayer. In the wheat field, the researchers found that the daily operation efficiency of the boom sprayer with a centrifugal sprinkler and hydraulic sprinkler was 20.87–29.56 ha·d$^{-1}$ [21]. Sun found that under the B5 (operation speed 4 m·s$^{-1}$, operation height 2 m, spraying flow 2 L·min$^{-1}$) and B9 (operation speed 3 m·s$^{-1}$, operation height 1.5 m, spraying flow 2 L·min$^{-1}$) parameters, the operation efficiency of the CE20 electric single-rotor plant protection UAV was 36.8 ha·d$^{-1}$ and 26.02 ha·d$^{-1}$, respectively [22]. After the application of the chemical topping agent, the growth rate of the cotton plant height gradually decreased and the plants almost stopped growing 20 days after being sprayed, which may be due to the inhibition of the development of the cotton top growing point [23]. After the application of the chemical topping agent by the T30 UAV, the growth rate of the cotton plant height was lower than that of the boom sprayer, and it could effectively control the plant type of cotton, effectively promote the transition from vegetative growth to reproductive growth, and increase the ratio of harvested bolls. After the application of the chemical topping agent, the length of the top first and second fruit branches and the internode length of the main stem of cotton were also relatively affected, their growth was slow, and the overall plant type of the cotton was not changed. The cotton petiole internode ratio is one of the indicators used to measure cotton growth [24]. This is affected by the cotton variety and field management measures used and can be used to reflect and predict cotton growth in the next stage. When the ratio of petiole length to the internode length of fruit branches in the same node is 1, the cotton is in the rapid growth stage; when the ratio of petiole length to the internode length of the fruit branches in the same node is 1.6, the cotton has a rapid growth trend; and when the ratio of petiole length to the internode length of the fruit branch in the same node is 2.5–3, the cotton is in a stable growth stage. In this study, the petiole internode ratio of the two varieties of cotton decreased first and then increased after pesticide spraying. Before pesticide spraying, the petiole internode ratio was about 1.6, with a trend of rapid growth. This may be due to the fact that irrigation was carried out within 10 days of spraying the plants with the chemical toping agent, which further reduced the stem node ratio. With the gradual decrease in the cotton growth rate, the petiole internode ratio gradually increased and slightly changed after 30 days of pesticide spraying. There is a phenomenon of interaction between agronomic properties, yield, and the quality traits of cotton. Among these, the number of bolls, single boll weight, and lint percentage are important indicators of cotton yield composition. The average length of the upper half, breaking specific strength, and micronaire value are important indicators of cotton fiber

quality composition [25]. At the same time, improving cotton yield and fiber quality are goals that researchers are committed to achieving. Compared with the boom sprayer, the spraying of a chemical topping agent with T30 UAV can effectively increase the boll number, upper half mean length, uniformity index, breaking specific strength, and elongation at break; reduce the micronaire value; and improve cotton quality. Plant height and uniformity index, elongation at break, number of bolls, lint percentage, single boll weight, micronaire value, and breaking specific strength were found to be highly significant and positively correlated. The upper half mean length showed a significant negative correlation with the micronaire value [26], which is consistent with the results obtained in our study. This shows that spraying a chemical topping agent has certain advantages in terms of improving the pesticide utilization rate, improving cotton agronomic properties, and increasing the yield and fiber quality. The use of a high-concentration pesticide solution with the T30 UAV will not cause phytotoxicity.

## 5. Conclusions

This study described the differences and similarities between two methods of spraying chemical topping agents and evaluated the operational effects of spraying a chemical topping agent using a T30 UAV and a boom sprayer. Through the comparison of two local planting varieties and the analysis of the results, the following conclusions can be drawn:

(1) For spraying a cotton chemical topping agent using a T30 UAV, the parameters should be set at a flight altitude of 2 m, a flight speed of 5 m·s$^{-1}$, a spray width of 5 m, and a spraying volume of 18 L ha$^{-1}$;

(2) After spraying using T30 UAV spraying, the plant height of cotton decreased by 1.46–2.70 cm compared with spraying using a boom sprayer, while the top first fruit branch inhibition rate was increased by 18.76% compared with spraying using a boom spraye;

(3) In terms of yield and fiber quality traits, the application of a chemical topping agent by T30 UAV could effectively improve the number of bolls per plant, upper half mean length, uniformity index, breaking specific strength, and elongation at break and reduce the micronaire value, thus improving the quality of cotton. Spraying with the T30 UAV had a significant effect on improving the utilization rate of pesticides and improving the agronomic properties, yield, and fiber quality of cotton.

**Author Contributions:** X.H. and Z.D. conceived and designed the experiments; Z.F., Y.L., Z.D. and L.D. performed the field experiments; Z.D. performed the chemical analysis; Z.D. and Z.F. analyzed the data; X.H wrote the paper; M.Z. and M.A. revised the manuscript. All authors have read and agreed to the published version of the manuscript.

**Funding:** This research was supported by the National Natural Science Foundation of China (31960566) and China Agriculture Research System (CARS-15-22).

**Acknowledgments:** The authors are grateful to SZ DJI Technology Co., Ltd., for providing plant protection unmanned aerial vehicle and related materials.

**Conflicts of Interest:** The authors declare no conflict of interest.

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
