# Peer review of "Comparison of the Effects of Chemical Topping Agent Sprayed by a UAV and a Boom Sprayer on Cotton Growth"

_agronomy, doi:10.3390/agronomy12071625_

Round 1

Reviewer 1 Report

General comments

This papers deals with the comparison of a ground based sprayer with UAV sprayer on coton crop. Deposition on crop and further quality criteria up to to teh yield of coton crop are studied.

Although well written, this paper really lacks of technical details as none of spraying systems are really described: application rates (only 18L/ha indicated for the UAV), type of nozzle, pressure, travel speed, etc…), spray swath… As a result it is difficult to compare results on a similar basis if application rates are different, or droplet size are radically different, etc.  

Many criteria between the two field plots (Fig 3 – to Fig 7) could also be due to agronomical or climatic conditions (not discussed) rather than the sprayer performance itself?

This last part also takes much place in the text and is not really deeply analysed (no statistical details) and with regards to the yield quantity and quality.

Detailed comments

Page 2 line 67 : this sentence shall be placed prior Line 61 where “twice applications” are introduced.

Page 2 lines 80-88 : Do the plots are different in fertility, soil type or rainfall that could also interfere with the results ?

Page 3 lines 91 – 104 : This paragraph shall contain more detailed information in order to be able to compare application equipments : nozzle types, flowrate, pressure, application rates, travel speed, etc.

Page 3 line 94 : I guess these dimensions correspond to the whole UAV?

Page 3 line 99: L/hm² even if absolutely correct, the paper also contains other units like l/ha so a harmonization to the practical unit (L/ha is suggested).

Page 3 line 102 : these dimensions correspond to the tank alone. Is this information fundamental?

Page 4 line 119 : g per hectare are used here . Please harmonize units among the text.

Page 6 line 164-165 : recovery rate for boom sprayer refers to which application rate ?

Page 7 Fig 3 : Is there a significant statistical difference between the 3 modalities?

Page 7 Fig 4 : Is there a significant statistical difference between the 2 modalities?

Page 10 Table 4 : are there differences between the 2 modalities?

Page 11 table 5 : many abreviations are not explained here… Since this table is the only source of statistical data between modalities, it is suggested to present those last data right after the deposition data and to interprete all previous criteria with regards to these quantitative and qualitative yield factors?

Author Response

 Dear Reviewer,

We are so grateful for the excellent suggestions and the detailed revising from you. We have benefited greatly from the revision. Now we response all the questions point by point and made modification accordingly.

Kind regards,

26th 6, 2022

This paper deals with the comparison of a ground based sprayer with UAV sprayer on coton crop. Deposition on crop and further quality criteria up to to teh yield of coton crop are studied.

Although well written, this paper really lacks of technical details as none of spraying systems are really described: application rates (only 18L/ha indicated for the UAV), type of nozzle, pressure, travel speed, etc…), spray swath… As a result it is difficult to compare results on a similar basis if application rates are different, or droplet size are radically different, etc.  

Response: Thank you for your kindly suggestions. We have added relevant technical details.

Many criteria between the two field plots (Fig 3 – to Fig 7) could also be due to agronomical or climatic conditions (not discussed) rather than the sprayer performance itself?

Response: Thank you for your kindly suggestions. In the two field plots, we evaluated the effect of cotton topping agent sprayed by UAV and boom sprayer at the same time. As far as a single plot is concerned, there are no agronomic or climatic problems.

This last part also takes much place in the text and is not really deeply analysed (no statistical details) and with regards to the yield quantity and quality.

Response: Thank you for your kindly suggestions. We further analyzed and discussed the data.

 Page 2 line 67 : this sentence shall be placed prior Line 61 where “twice applications” are introduced.

Response: Thank you for your kindly suggestions. We had revised it.

Page 2 lines 80-88 : Do the plots are different in fertility, soil type or rainfall that could also interfere with the results ?

Response: Thank you for your kindly suggestions. Drip irrigation with one film and two tubes was used during whole growth period. The planting mode, growth period, fertilization level, irrigation and conventional chemical topping agent spraying time and dose of each treatment were the same. At the same time, we chose a larger plot area to avoid the error of test results caused by the inconsistency of local water and fertilizer conditions.

Page 3 lines 91 – 104 : This paragraph shall contain more detailed information in order to be able to compare application equipments : nozzle types, flowrate, pressure, application rates, travel speed, etc.

Response: Thank you for your kindly suggestions. We had supplied these parameters.

Page 3 line 94 : I guess these dimensions correspond to the whole UAV?

Response: Yes, this is the size of the whole UAV when it is fully deployed.

Page 3 line 99: L/hm² even if absolutely correct, the paper also contains other units like l/ha so a harmonization to the practical unit (L/ha is suggested).

Response: Thank you for your kindly suggestions. We had revised the full text.

Page 3 line 102 : these dimensions correspond to the tank alone. Is this information fundamental?

Response: Yes, this is the size of the tank alone. The length of boom after spreading is 12 m.

Page 4 line 119 : g per hectare are used here . Please harmonize units among the text.

Response: Thank you for your kindly suggestions. We had revised it.

Page 6 line 164-165 : recovery rate for boom sprayer refers to which application rate ?

Response: The pesticide utilization rate was calculated by allure red tracer.

Page 7 Fig 3 : Is there a significant statistical difference between the 3 modalities?

Response: We have added variance analysis to the figure.

Page 7 Fig 4 : Is there a significant statistical difference between the 2 modalities?

Response: We have added variance analysis to the figure.

Page 10 Table 4 : are there differences between the 2 modalities?

Response: We have added variance analysis to the table.

Page 11 table 5 : many abreviations are not explained here… Since this table is the only source of statistical data between modalities, it is suggested to present those last data right after the deposition data and to interprete all previous criteria with regards to these quantitative and qualitative yield factors?

Response: I am very sorry for the mistake. We have added the complete spelling of abbreviations and added the discussion.

Reviewer 2 Report

Dear Authors,

below is a list of remarks and suggestions to improve the manuscript.

Detailed remarks:

 Please revise in the document all the measurement units and their writing. Use the form xxx-1

Line 78: insert an image and a zoom of the study area following the journal’s instructions

Line 80: When do you perform the test? Please provide additional information on the date, and timing.

Line 80: Provide additional information on wheatear condition

Line 87-88: Provide specific details for each practice. Which was the plant variability in terms of height? Which were the soil properties? Which was the orientation? Which was the slope?

Line 92 please correct as follows g L -1

Line 99 How do you measure the spray amplitude? Moreover, add the path planning

Line 103 please correct the highlighted in yellow

Line 103 Provides additional essential information related to the boom sprayer: The forwarding speed, the boom height from the ground, the nozzle model, and the manufacturer the working pressure.

Line 103 do you have check the flow rate before the spraying?

Table 3: Add a column with the tools used or in the line 148 explain the measurements techniques

Line 153 how do you measure weight and yield? Instruments? Techniques?

Line 155 How does Urumqi perform the analysis?

Line 307 Discussions are too general, please analyze more in deep your results

Line 311

Your statement: Among two sprayers, T30 UAV has less pesticide loss to the ground and more deposition in each canopy of cotton [19].

How do you evaluate the pesticide loss to the ground?

Please refer the discussion on the achievement of your results on lines 312 to 317

I suggest to follow the material and method sections of those articles where you can find specific details in light of the similarity of the topic:

Sarri, D., Martelloni, L., Rimediotti, M., Lisci, R., Lombardo, S., & Vieri, M. (2019). Testing a multi-rotor unmanned aerial vehicle for spray application in high slope terraced vineyard. Journal of Agricultural Engineering, 50(1), 38-47. doi:10.4081/jae.2019.853

Santana, L. S., Ferraz, G. A. E. S., Cunha, J. P. B., Santana, M. S., Faria, R. O., Marin, D. B., . . . Sarri, D. (2021). Monitoring errors of semi-mechanized coffee planting by remotely piloted aircraft. Agronomy, 11(6) doi:10.3390/agronomy11061224

I am wondering that not a single word has been included concerning the normalizing data because the authors compared quite different volume rates (40 versus 1.2 L on 667 m-2). The Authors have to transform values to make a clear comparison to published literature values.

I suggest using different words than the title in order to improve the research in databases.

The abstract exceeds the maximum limit of 200 imposed by the Agronomy journal please correct

Author Response

Dear Reviewer,

We are so grateful for the excellent suggestions and the detailed revising from you. We have benefited greatly from the revision. Now we response all the questions point by point and made modification accordingly.

Kind regards,

26th 6, 2022

Please revise in the document all the measurement units and their writing. Use the form xxx-1

Response: Thank you for your kindly suggestions. We had revised the full text.

Line 78: insert an image and a zoom of the study area following the journal’s instructions

Response: Thank you for your kindly suggestions. We had added the figure which including zoom of the study.

Line 80: When do you perform the test? Please provide additional information on the date, and timing.

Response: The spraying date of topping agent was shown in Table 1. We also make a further supplement in the manuscript.

Line 80: Provide additional information on wheatear condition

Response: We had supplemented the weather information at the time of spraying.

Line 87-88: Provide specific details for each practice. Which was the plant variability in terms of height? Which were the soil properties? Which was the orientation? Which was the slope?

Response: We had supplemented the schematic diagram of the test site, in which relevant information was marked (Figure 1). Drip irrigation with one film and two tubes was used during whole growth period. The planting mode, growth period, fertilization level, irrigation and conventional chemical topping agent spraying time and dosase of each treatment were the same. Test site a extends from north to south, with an altitude of 362 m in the north and 361 m in the south. The test site b extends from northwest to southeast, with an altitude of 343 m in the northwest and 342 m in the southeast. The altitude difference between the two test sites did not exceed 1 m.

Line 99 How do you measure the spray amplitude? Moreover, add the path planning

Response: When planning the UAV route, set the route spacing by the remote controller to make each UAV route interval 5m (Figure 2). The spray amplitude has been changed to route spacing.

Line 103 please correct the highlighted in yellow

Response: Thank you for your kindly suggestions. We had revised the full text.

Line 103 Provides additional essential information related to the boom sprayer: The forwarding speed, the boom height from the ground, the nozzle model, and the manufacturer the working pressure.

Response: We had supplemented the parameter. The forwarding speed was 4.2~4.5 km h-1, the boom height from the ground was 60cm, the nozzle model was Fan-shaped nozzle, and the working pressure was 0.2~0.4 Mpa.

Line 103 do you have check the flow rate before the spraying?

Response: After measuring the nozzle flowrate at the left, middle and right positions of the boom for three times, the flowrate of a single nozzle was 1.37 L min-1.

Table 3: Add a column with the tools used or in the line 148 explain the measurements techniques

Response: We had supplemented the information. Cotton plant height, number of fruit branches, petiole internode ratio, fruit branch length (top first and top second), internode length (the internode length of the top first to the top second fruit branches) measured with pocket ruler (Deli Group Co., Ltd., Ningbo, China), 1 day before and after 10, 20, 30 and 40 days of pesticide spraying.

Line 153 how do you measure weight and yield? Instruments? Techniques?

Response: I am very sorry for the mistake. We had added the method in the new manuscript.

Line 155 How does Urumqi perform the analysis?

Response: We added the instrument information for cotton quality determination.

Line 307 Discussions are too general, please analyze more in deep your results.

Response: Thank you for your kindly suggestions. Combined with the experimental data and relevant references, we made an in-depth revision to the Discussions.

Line 311 Your statement: Among two sprayers, T30 UAV has less pesticide loss to the ground and more deposition in each canopy of cotton [19].

How do you evaluate the pesticide loss to the ground?

Response: Thank you for your kindly suggestions. We didn't measure the pesticide loss on the ground. I think this conclusion is a little hasty. We deleted this sentence.

Please refer the discussion on the achievement of your results on lines 312 to 317

I suggest to follow the material and method sections of those articles where you can find specific details in light of the similarity of the topic:

Sarri, D., Martelloni, L., Rimediotti, M., Lisci, R., Lombardo, S., & Vieri, M. (2019). Testing a multi-rotor unmanned aerial vehicle for spray application in high slope terraced vineyard. Journal of Agricultural Engineering, 50(1), 38-47. doi:10.4081/jae.2019.853

Santana, L. S., Ferraz, G. A. E. S., Cunha, J. P. B., Santana, M. S., Faria, R. O., Marin, D. B., . . . Sarri, D. (2021). Monitoring errors of semi-mechanized coffee planting by remotely piloted aircraft. Agronomy, 11(6) doi:10.3390/agronomy11061224

Response: Thank you for your kindly suggestions. We had modified those sentences.

I am wondering that not a single word has been included concerning the normalizing data because the authors compared quite different volume rates (40 versus 1.2 L on 667 m-2). The Authors have to transform values to make a clear comparison to published literature values.

Response: Thank you for your kindly suggestions.

I suggest using different words than the title in order to improve the research in databases.

Response: Thank you for your kindly suggestions.

The abstract exceeds the maximum limit of 200 imposed by the Agronomy journal please correct

Response: We had modified the abstract, and the number of words had met the requirements of Agronomy.

Round 2

Reviewer 1 Report

The paper was significantly improved with consideration to reviewer's comments.

Author Response

 Dear Reviewer,

We are so grateful for the excellent suggestions and the detailed revising from you. We have benefited greatly from the revision. Now we response all the questions point by point and made modification accordingly.

Kind regards,

1th, 7, 2022

I don't feel qualified to judge about the English language and style.

Response: The English language and style of manuscript had been edited by MDPI.

The paper was significantly improved with consideration to reviewer's comments.

Response: Thank you for your kindly comments.

Reviewer 2 Report

Dear Authors,

I suggest the following additional remarks and suggestions:

line 90 please insert a graphical map of the study area

line 125 (and in all the document use m s-1)

line 126 brand of the nozzle and manufacturer, did you use an ISO nozzle?

Line 144 Authors stated: "The average plant height of cotton in test site a was 52 cm, and the average plant height of cotton 145 in test site b was 70 cm. I wondering how did you were able to insert this plant in a polyethene bag of 28x38cm. I guess that there is something wrong, please check.

Line 66-70 I suggest moving the paragraph at line 400 to line 66-77 where fit better. I think that you have to highlight that UAVs have been used profitably also in other scenarios with good results.

Author Response

Dear Reviewer,

We are so grateful for the excellent suggestions and the detailed revising from you. We have benefited greatly from the revision. Now we response all the questions point by point and made modification accordingly.

Kind regards,

1th, 7, 2022

line 90 please insert a graphical map of the study area

Response: Thank you for your kindly suggestions. We had inserted a graphical map of the study area.

line 125 (and in all the document use m s-1)

Response: Thank you for your kindly suggestions. We had revised the full text.

line 126 brand of the nozzle and manufacturer, did you use an ISO nozzle?

Response: The nozzle is produced by Dongguan Huajue Spray Technology Co., Ltd., we had added the information in the manuscript.

Line 144 Authors stated: "The average plant height of cotton in test site a was 52 cm, and the average plant height of cotton 145 in test site b was 70 cm. I wondering how did you were able to insert this plant in a polyethene bag of 28x38cm. I guess that there is something wrong, please check.

Response: We used scissors to cut the cotton plant into sections so that they could be put into a polyethylene bag (28 cm × 38 cm).

Line 66-70 I suggest moving the paragraph at line 400 to line 66-77 where fit better. I think that you have to highlight that UAVs have been used profitably also in other scenarios with good results.

Response: Thank you for your kindly suggestions. We had revised that paragraph.